**Data Availability Statement:** All relevant data are within the manuscript and its Supporting information files.

# Longitudinal changes in anthropometric, physiological, and physical qualities of international women's rugby league players

Sean Scantlebury[1,2]*, Nessan Costello[1], Cameron Owen[1,2], Sarah Chantler[1,2], Carlos Ramirez[1], Santiago Zabaloy[3], Neil Collins[1,2], Hayden Allen[1], Gemma Phillips[2,4], Marina Alexander[1], Matthew Barlow[1], Emily Williams[1], Peter Mackreth[1], Stuart Barrow[2], Parag Parelkar[1], Anthony Clarke[1], Benjamin Samuels[1], Stephanie Roe[1], Cameron Blake[1], Ben Jones[1,2,5,6,7]

1 Carnegie School of Sports, Leeds Beckett University, Leeds, United Kingdom, 2 England Performance Unit, Rugby Football League, Manchester, United Kingdom, 3 Faculty of Physical Activity and Sports, University of Flores, Buenos Aires, Argentina, 4 Hull Kingston Rovers, Hull, United Kingdom, 5 Division of Physiological Sciences and Health through Physical Activity, Lifestyle and Sport Research Centre, Department of Human Biology, Faculty of Health Sciences, University of Cape Town, Cape Town, South Africa, 6 School of Behavioural and Health Sciences, Faculty of Health Sciences, Australian Catholic University, Brisbane, Queensland, Australia, 7 Premiership Rugby, London, United Kingdom

* s.scantlebury@leedsbeckett.ac.uk

## Abstract

This is the first study to assess longitudinal changes in anthropometric, physiological, and physical qualities of international women's rugby league players. Thirteen forwards and 11 backs were tested three times over a 10-month period. Assessments included: standing height and body mass, body composition measured by dual x-ray absorptiometry (DXA), a blood panel, resting metabolic rate (RMR) assessed by indirect calorimetry, aerobic capacity (i.e., $\dot{V}O_2$max) evaluated by an incremental treadmill test, and isometric force production measured by a force plate. During the pre-season phase, lean mass increased significantly by ~2% for backs (testing point 1: 47 kg; testing point 2: 48 kg) and forwards (testing point 1: 50 kg; testing point 2: 51 kg) (p = ≤ 0.05). Backs significantly increased their $\dot{V}O_2$max by 22% from testing point 1 (40 ml kg$^{-1}$ min$^{-1}$) to testing point 3 (49 ml kg$^{-1}$ min$^{-1}$) (p = ≤ 0.04). The $\dot{V}O_2$max of forwards increased by 10% from testing point 1 (41 ml kg$^{-1}$ min$^{-1}$) to testing point 3 (45 ml kg$^{-1}$ min$^{-1}$), however this change was not significant (p = ≥ 0.05). Body mass (values represent the range of means across the three testing points) (backs: 68 kg; forwards: 77–78 kg), fat mass percentage (backs: 25–26%; forwards: 30–31%), resting metabolic rate (backs: 7 MJ day$^{-1}$; forwards: 7 MJ day$^{-1}$), isometric mid-thigh pull (backs: 2106–2180 N; forwards: 2155–2241 N), isometric bench press (backs: 799–822 N; forwards: 999–1024 N), isometric prone row (backs: 625–628 N; forwards: 667–678 N) and bloods (backs: ferritin 21–29 ug/L, haemoglobin 137–140 g/L, iron 17–21 umol/L, transferrin 3 g/L, transferring saturation 23–28%; forwards: ferritin 31–33 ug/L, haemoglobin 141–145 g/L, iron 20–23 umol/L, transferrin 3 g/L, transferrin saturation 26–31%) did not change (p = ≥ 0.05). This study provides novel longitudinal data which can be used to better prepare

**Funding:** The author(s) received no specific funding for this work.

**Competing interests:** "I have read the journal's policy and the authors of this manuscript have the following competing interests: SS is the strength and conditioning coach for England women's rugby league SC is the nutritionist for England women's rugby league NC is the sports scientist for England women's rugby league GP is the head of medical for England women's rugby league SB is the manager of England women's rugby league. BJ is the head of performance for England rugby league" This does not alter our adherence to PLOS ONE policies on sharing data and materials.

women rugby league players for the unique demands of their sport, underpinning female athlete health.

## Introduction

Rugby league is a field-based team sport characterised by intermittent bouts of high-intensity actions such as sprinting, changes of direction, tackling and collisions interspersed by bouts of low-intensity activity [1–3]. These demands necessitate the development of specific physical attributes to optimise performance, support recovery, and reduce injury risk [4–6]. International women's rugby league players have superior speed, power, and high-intensity intermittent running capabilities compared to domestic players [7]. While previous studies have reported the anthropometric and physical characteristics of international women's rugby league players [7–9] longitudinal changes in anthropometric and physical qualities have not been reported.

Longitudinal changes in anthropometric and physical qualities are important, to evaluate how women's rugby league players develop within their unique context. For example, access to training facilities and expert guidance, as well as player's work commitments, can restrict training opportunities, thereby influencing their long-term physical development [10]. As such, there is a need to increase the evidence base in women's rugby league [11], aligning with the identified research priority of physical performance in women's rugby [12].

The longitudinal quantification of elite women's rugby league players anthropometric, physiological, and physical attributes are needed to establish normative standards for pre- and in-season periods. These standards can be used in athlete profiling, identifying strengths and weaknesses, performance monitoring, and talent identification strategies [13]. Therefore, the aim of this study is to use valid, and reliable testing measures to quantify changes in body composition, blood markers, energy requirements, strength, and aerobic capacity of women's international rugby league players over three testing points in the lead up to the 2022 Rugby League World Cup (RLWC).

## Methods

### Experimental approach to the problem

This study utilised a cohort study design to establish the anthropometric, physiological, and physical characteristics of international women's rugby league players leading up to the 2022 RLWC. To achieve this, participants completed testing at three testing points; 1) December 2021: The beginning of the 2022 Women's Super League (WSL) pre-season, 2) March 2022: the end of the 2022 WSL pre-season, and 3) October 2022: following the completion of the 2022 WSL season and prior to the RLWC.

Data collection included anthropometric measurements (height and body mass), body composition, resting metabolic rate (RMR), blood profiling, alongside assessments of endurance capacity (maximum oxygen consumption [$\dot{V}O_2max$]) and isometric strength.

Body composition and RMR were measured by dual-energy x-ray absorptiometry (DXA) and indirect calorimetry, respectively. Aerobic capacity was assessed through an incremental treadmill protocol, while isometric force production involved measurements of the isometric mid-thigh pull, isometric bench press, and isometric prone row via force plates.

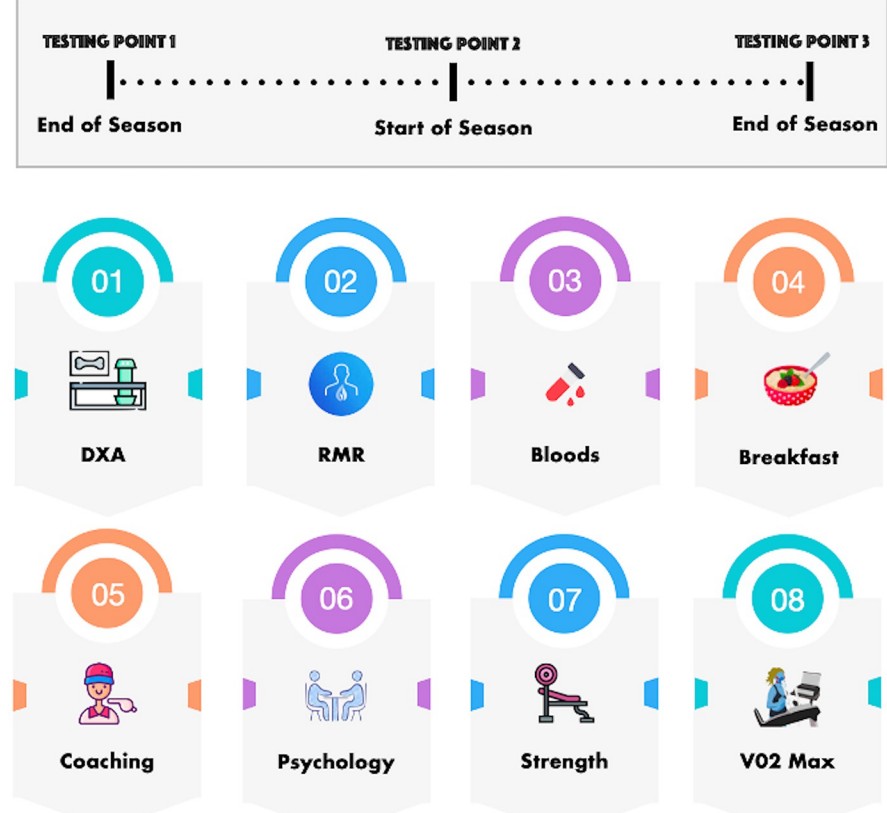

**Fig 1. A typical testing day schedule.** Dual-energy x-ray absorptiometry (DXA). Resting metabolic rate (RMR).

A visual representation of a typical testing day is shown in Fig 1. Participants fasted until they completed the RMR assessment, DXA scan, and blood tests. Subsequently, participants were provided with a standardised breakfast consisting of overnight oats with chia seeds, stewed apple, and granola. This breakfast provided approximately 524 kilocalories per portion, with a macronutrient composition of 55 g carbohydrate, 10 g of protein, and 26 g of fat. Participants consumed fluids *ab libitum*. Following breakfast, participants engaged in individual coaching and psychology workshops.

## Participants

The sample size for this study was determined by the maximum squad size permitted for the 2022 RLWC. Consequently, a total of 24 women's rugby league players were purposefully recruited. Recruitment began on the 7th of December 2021 and ended on the 27th of October 2022. Participants provided written informed consent. Participants were categorised as Tier 4 "elite/international level" [14]. To be eligible for participation at each testing point, participants were required to meet the following criteria: 1) international squad registration, 2) >18 years, and 3) absence of injury or illness on the day of testing. Therefore, the final sample comprised of 24 international women's rugby league players, with 13 classified as forwards (age: 26.2 ± 4.6 years; height: 167.4 ± 4.1 cm), and 11 classified as backs (age: 26.3 ± 5.8 years; height: 169.8 ± 4.6 cm). Not all participants completed all tests at each testing point (Table 1).

**Table 1. The number of participants who completed each test at each testing point.**

| Test | Testing point 1 (n) | Testing point 2 (n) | Testing point 3 (n) |
|---|---|---|---|
| Height | 17 | 21 | 24 |
| Body Mass | 17 | 21 | 24 |
| Blood work | 17 | 21 | 24 |
| Dual-energy x-ray absorptiometry | 17 | 21 | 24 |
| Resting metabolic rate | 17 | 21 | 24 |
| Isometric mid-thigh pull | 17 | 19 | 24 |
| Isometric bench press | - | 19 | 24 |
| Isometric prone row | - | 19 | 24 |
| $\dot{V}O_2max$ | 13 | 15 | 21 |

Participants competed in the highest standard of the domestic women's rugby league competition (i.e., WSL), where they regularly engaged in two to three training sessions and one match per week during the season, in addition to international training. Throughout the season, participants followed physical training programmes prescribed by the international strength and conditioning coach at their respective domestic clubs. Further details regarding their annual training plan are provided in Table 2, and representative pre- and in-season weekly training microcycles are outlined in Tables 3 and 4. Weekly training schedules varied among participants due to their club training, competition, and work schedules. The research ethics committee of Leeds Beckett University gave ethical approval for this work (reference number: 91130).

**Table 2. Summary of the macrocycle for an international women's rugby league squad.**

| Stage of the Season | Training Phase | Duration | Main Objectives |
|---|---|---|---|
| Off-Season | General preparation and conditioning | October 2021 –January 2022 | Maintain physical qualities (e.g., speed, endurance, and strength) and recover from rugby specific training and competition. |
| | | Frequency: 2 to 3 sessions a week. | |
| Pre-Season | Specific preparation and pre-competitive | January 2022 –March 2022 | Develop physical qualities (i.e., speed, endurance, and strength). Strength training sessions focused on muscular hypertrophy. Speed development was incorporated into pre-training session warmups. |
| | | Frequency: 4 to 5 sessions a week. | Endurance capacity was developed using small sided games. Conditioning sessions were prescribed via the individuals' end velocity following the 30:15 intermittent fitness test [15]. |
| In-Season | Competitive | March 2022 –September 2022 | Continue to develop physical qualities (i.e., speed, endurance, strength, and power). Strength training sessions included max strength, strength-speed, and plyometric exercises. Speed development was incorporated into pre-training session warmups. |
| | | Frequency: 3 to 4 sessions a week. | |
| | | 1 weekly competition (friendly and/or official) | Conditioning sessions were prescribed via the individual's end velocity following the 30:15 intermittent fitness test. |
| Pre-World Cup | Specific preparation and pre-competitive | September 2022 –October 2022 | Continue to develop physical qualities (i.e., speed, endurance, strength, and power). Strength training sessions included max strength, strength-speed, and plyometric exercises. Speed development was incorporated into pre-training session warmups. |
| | | Frequency: 3 to 4 sessions a week. | Conditioning sessions were prescribed via the individual's end velocity following the 30:15 intermittent fitness test. |

**Table 3. Typical pre-season weekly training schedule of an international women's rugby league squad.**

| Rest | Conditioning (30') (80% VIFT) 3 mins work, 2 mins rest 1 set x 6 reps | TEC/TAC (60'-120') *Speed and conditioning included in tec/tac.* **Speed block** (10–20') Sprint mechanics Acceleration 10m to 20m (2 reps) Max velocity 40m (1–3 reps) **Conditioning** (20') (90%-100% VIFT) 15s -30s work, 15s–30s rest 2–3 sets x 8–10 reps **Strength Block** (60'): Hypertrophy Lower body Traditional exercises included (but not limited to) during the mesocycle: BS, FS, TBD, RDL, Nordics, BSS, LL, HT, CR, PR, plank, BR. (70%–80% 1RM) | Rest | TEC/TAC (60'-120') *Speed and conditioning included in tec/tac.* **Speed block** (10–20') Sprint mechanics Acceleration 10m–20m (2 reps) Max velocity 40m (1–3 reps) **Conditioning** (20') (90%-100% VIFT) 15s -30s work, 15s–30s rest 2–3 sets x 8–10 reps **Strength Block** (60') Hypertrophy Upper body Traditional exercises included (but not limited to) during the mesocycle: BP, SA DB row, SA DB press; Pull-ups, SA DB SP, BC, TE, PR, Dead bugs. (70%–80% 1RM) | Rest | TEC/TAC (60'-120') *Speed and conditioning included in tec/tac.* **Speed block** (10–20') Sprint mechanics Acceleration 10m–20m (2 reps) Max velocity 40m (1–3 reps) **Conditioning** (20') (90%-100% VIFT) 15s -30s work, 15s–30s rest 2–3 sets x 8–10 reps **Strength Block** (60') Hypertrophy Full body Traditional exercises included (but not limited to) during the mesocycle: BS, FS, RDL, Nordics, BSS, LL, HT, CR BP, SA DB row, SA DB press, SA DB SP, MP, Pull ups, Plank, Dead bugs. (70%–80% 1RM) |
|---|---|---|---|---|---|---|

Abbreviations: TEC/TAC, technical and tactical training; VIFT, 30:15 intermittent fitness test end speed; 1RM, 1-repetition maximum; SA, single arm; DB, dumbbell; BS, back squat; FS, front squat; TBD, trap bar deadlift; RDL, Romanian deadlift; BSS, Bulgarian split squat; LL, lateral lunge; HT, hip thrust; CR, calf raise; BP, bench press; BO row, bent-over row; SP, shoulder press; MP, military press; BC, bicep curl; TE, tricep extension; PR, Paloff rotation; BR, barbell rollout.

**Table 4. Typical in-season weekly training schedule of an international women's rugby league squad.**

| Sunday | Monday | Tuesday | Wednesday | Thursday | Friday | Saturday |
|---|---|---|---|---|---|---|
| **Competition** | Rest | **TEC/TAC** (60'-120') *Speed and conditioning included in tec/tac.* **Speed block** (10–20') Sprint mechanics Acceleration 10m to 20m (2 reps) Max velocity 40m (1–3 reps) **Conditioning** (20') (90%-100% VIFT) 15s–30s work, 15s–30s rest 2–3 sets x 8–10 reps **Strength Block** (60'): S/PT Lower body Traditional exercises included (but not limited to) during the mesocycle: JS, DJ, BJ, Bounding, BS, FS, TBD, RDL, Nordics, BSS, LL, HT, CR, PR, plank, BR. (70%–90% 1RM) | **Strength Block** (60'): S/PT Upper body Traditional exercises included (but not limited to) during the mesocycle: HC, HP, PPU, SA DB row, SA DB press; Pull-ups, SA DB SP, BC, TE, PR, Dead bugs. (70%–90% 1RM) | **TEC/TAC** (60'-120') *Speed and conditioning included in tec/tac.* **Speed block** (10–20') Sprint mechanics Acceleration 10-m–20m (2 reps) Max velocity 40m (1–3 reps) **Conditioning** (20') (90%-100% VIFT) 15s -30s work, 15s–30s rest 2–3 sets x 8–10 reps | **Strength Block** (60'): S/PT Full body Traditional exercises included (but not limited to) during the mesocycle: HP, Jump squat, BJ, bounding, MBS, deadlift, TBD, BP, BO row, SP, pull-ups, Nordics, Isometric hamstring hold, PR. (70%–90% 1RM) | Rest |

Abbreviations: TEC/TAC, technical and tactical training; VIFT, 30:15 intermittent fitness test end speed; 1RM, 1-repetition maximum; S/PT, strength/power training; SA, single arm; DB, dumbbell; HC, hang clean; HP, high pulls; JS, jump shrug; DJ, drop jump; BJ, box jump; MBS, med ball slam; PPU, plyometric push-up; BS, back squat; FS, front squat; TBD, trap bar deadlift; RDL, Romanian deadlift; BSS, Bulgarian split squat; LL, lateral lunge; HT, hip thrust; CR, calf raise; BP, bench press; BO row, bent-over row; SP, shoulder press; LP, landmine press; BC, bicep curl; TE, tricep extension; PR, Paloff rotation; BR, barbell rollout.

## Procedures

**Anthropometrics and body composition.** On the morning of testing, measurements of fasted body mass (SECA, model-875, Hamburg, Germany) and stature (SECA, model-213, Hamburg, Germany) were taken to the nearest 0.1 kg and 0.1 cm, following the guidelines of the International Society for the Advancement of Kinanthropometry.

Participants then completed a DXA scan (Lunar iDXA, GE Medical Systems, United Kingdom) following established best practice protocols [3]. The scans were conducted with participants in a euhydrated state (osmolality: <700 mOsmol·kg$^{-1}$), wearing minimal clothing, and with shoes and jewellery removed. To minimise potential confounding factors, participants refrained from intensive exercise, consuming alcohol, or consuming caffeine for ≥8 hours prior the scan.

The GE Lunar iDXA system (GE Healthcare, Pollards Wood, Buckinghamshire; Encore software version 18) employed the OneScan functionality for performing lumbar spine (L1-L4) and dual hip (total hip and neck of femur) scans without repositioning the participants. Correct positioning was ensured by utilising the GE dual femur positioning device, which required participants to rotate both legs internally between 15 and 25 degrees. Total body positioning involved centring participants on the table, with their hands at their sides, and Velcro straps used to secure their legs at the ankles. Regular quality assurance was maintained through calibration block and spine phantom assessments, which showed no observed drifts.

In-vivo precision (coefficient of variation, CV) for the DXA measurements conducted by the Leeds Beckett DXA unit were as follows: 0.82% for lumbar spine bone mineral density (BMD), 0.98% for total hip BMD and 1.34% for femoral neck BMD. The CV for total body composition measurements were 0.99% for fat, 0.98% for fat mass, and 0.42% for lean.

**Resting metabolic rate.** Resting metabolic rate was measured using open-circuit indirect calorimetry (Cortex 3B-R3 MetaLyzer, CORTEX Biophysik GmbH, Leipzig, Germany) and a transparent ventilated hood system (Cortex Canopy System, CORTEX Biophysik GmbH, Leipzig, Germany). All measurements were conducted in accordance with established best practice protocols, including an overnight fast and abstention from alcohol, nicotine, and caffeine for a minimum of 8 hours [16].

Data were collected over 20 minutes, with the second 10 minutes used to calculate RMR [17], as previously described [18]. The average CV for $\dot{V}O2$ (oxygen consumption), $\dot{V}CO2$ (carbon dioxide production), and respiratory exchange ratio (RER) across the three testing points was 10.6 ± 3.7%, 11.6 ± 4.9%, and 6.4 ± 2.7%, respectively.

**Blood measures.** Venous blood samples were collected using standardised venepuncture techniques, performed by a qualified medical practitioner. Blood samples were drawn into appropriate tubes (BD Vacutainer ®, Plymouth) to conduct a full blood count and assess the iron profile of participants. For the iron profile analysis, blood samples were centrifuged at 3000 revolutions per minute for a duration of 10 minutes at a temperature of 4 degrees Celsius. Subsequently, the resulting serum was pipetted into microtubes and then stored at -20 degrees Celsius to preserve it for future analysis. All blood samples were analysed together at an external laboratory (Northumbria, UK). The assessment of serum iron, ferritin and transferrin levels was carried out using standardised procedures and manufactures instructions (AU Beckman Coulter analyser).

**Isometric force.** Prior to commencing the strength testing, which included the isometric mid-thigh pull, isometric bench-press, and isometric prone row, participants completed a standardised warm-up consisting of dynamic stretches and simple bodyweight movements

such as squats, lunges, push-ups. The warm-up was overseen by a qualified strength and conditioning coach and lasted 5 minutes.

The isometric mid-thigh pull, was selected as a measure of global full body strength [19], and executed in a specialised adjustable rack equipped with a force plate (Kistler [family type 9260AA]). Procedures of the isometric mid-thigh pull followed previously published guidelines [20]. The height of the bar was adjusted to align with each participant's mid-thigh level, resulting in a knee angle of 125°-145° and hip flexion angle of 140°-150°. Participants completed this test twice, with a 3 minute rest between efforts [21]. To elicit maximal effort, participants were instructed to exert maximum force as quickly as possible after a 3 second countdown [21]. The highest peak vertical force recorded from the two attempts was used and expressed in Newtons (N).

The isometric bench press and isometric prone row were performed using specialised adjustable racks connected to a force plate (Kistler [family type 9253B]) following previously published guidelines [22, 23]. In the isometric bench press, participants were positioned with their back flat on the bench fitted within the rack, ensuring that the bar was level with their chest. For the isometric prone row, participants lay with their chest flat on the bench within the rack, with the bar positioned at chest level. In both tests, participants were required to maintain a 90˚—120˚ elbow angle, and the bar position was adjusted accordingly. During both the isometric bench press and isometric prone row, participants were instructed to exert maximum force as rapidly as possible. Two attempts were recorded for each test, with a 3-minute rest period between trials. Verbal encouragement was provided during each attempt. The highest peak vertical force recorded from the two attempts was used and expressed in Newtons (N). Notably, the isometric bench press and isometric prone row were not completed during testing point 1 due to equipment failure.

**Aerobic capacity.** Maximal oxygen uptake was determined through a running-based incremental ramp exercise test conducted on a slat-belt treadmill (Woodway ELG, Woodway, Birmingham, UK). The test protocol began with 3 minutes of walking at 3 km/h, followed by an initial running speed of 7 km/h. The running speed increased by 1 km/h every minute until the participants reached volitional exhaustion. The treadmill was set to a 1% incline throughout the test.

Pulmonary gaseous exchange was assessed using online, breath-by-breath gas measurement (Metalyzer 3B; Cortex Medical, Germany). Prior to the test, a two-point calibration of the gas analyser was performed in accordance with the manufacturer's guidelines. The calibration of the $O_2$ and $CO_2$ analysers was executed using ambient air and a known calibration gas (15.00% $O_2$, 5.00% $CO_2$) (Cortex Medical, Germany). A 3 L calibration syringe (Hans Rudolph, Kansas, USA) was employed for the calibration of the volume transducer.

To measure heart rate responses during exercise, a Bluetooth heart rate strap (Polar H10; Polar, Finland) was utilised. The highest 30 second average $\dot{V}O_2$ value attained during the incremental ramp test was selected as the criterion for determining the participant's $\dot{V}O_{2max}$.

## Statistical analysis

All statistical analyses were conducted using RStudio (V 4.2.0, RStudio, Boston, MA, USA). Estimated means (*emmeans*) [24] were determined for each testing point using mixed models (*lme4*) [25]. The characteristic of interest served as the dependent variable, while testing point was included as a fixed effect, with player considered as a random effect. When examining differences between player positions (forwards and backs), position was added as a fixed effect, and an interaction term was incorporated into the model. Model assumptions were assessed

using the *performance* package [26]. An outlier exceeding 1200 N was removed from the prone row data prior to modelling.

Ferritin and transferrin saturation were identified as log normal in distribution; therefore, outcomes were log transformed before analysis, and the results were back-transformed for reporting. Body fat percentage was modelled using a beta distribution, considering its values range between 0 and 1. Post-hoc comparisons were made between player positions and testing timepoints, with a significance level set as alpha = 0.05. Means and 95% confidence intervals (95% CI) were reported, along with effect sizes and rate ratios where appropriate.

## Results

### Anthropometrics and body composition

Table 5 displays the mean (95% confidence intervals) for body mass, lean mass, fat mass, and body fat % for backs and forwards during the three testing points. Backs had a significantly lower body mass and body fat % than forwards across all three testing points (p = ≤ 0.04). Backs and forwards had a significant increase in lean mass from testing point 1 to testing point 2 (backs: Δ1.1 kg; forwards: Δ0.9 kg; p ≤ 0.02). There were no other significant changes in body composition across testing points.

### Energy requirements

The RMR (range of means across the three testing points) for backs (6.53–7.07 MJ.day$^{-1}$) and forwards (6.55–7.04 MJ.day$^{-1}$) did not significantly differ at testing points 1 (p = 0.87),

**Table 5. The mean (95% CI) for body composition, energy requirements, blood measures, endurance capacity, and isometric strength of backs and forwards across the three testing points.**

| | | Backs | | | Forward | | |
|---|---|---|---|---|---|---|---|
| | | **Testing point 1** | **Testing point 2** | **Testing point 3** | **Testing point 1** | **Testing point 2** | **Testing point 3** |
| **Body composition** | Body Mass (kg) | 68^ (62–73) | 68^ (63–74) | 68^ (63–74) | 77 (71–82) | 78 (72–83) | 77 (72–82) |
| | Lean Mass (g) | 47170* (43985–50354) | 48314 (45149–51478) | 47916 (44761–51071) | 50411* (4703–53320) | 51333 (48428–54238) | 50883 (47981–53784) |
| | Fat Mass (g) | 17405^ (13880–20930) | 17053^ (13593–20513) | 17127^ (13698–20555) | 22945 (19770–26119) | 23077 (19912.2–26241.5) | 22866 (19712–26019) |
| | Body Fat (%) | 26^ (24–20) | 25^ (23–28) | 26^ (23–29) | 31 (28–34) | 31 (28–33) | 30 (28–33) |
| **Energy Requirements** | RMR (MJ·day$^{-1}$) | 7.07 (6.44–7.70) | 6.61 (6.05–7.17) | 6.53 (6.04–7.02) | 7.01 (6.52–7.49) | 7.04 (6.57–7.51) | 6.55 (6.10–7.00) |
| **Blood work** | Ferritin (ug/L) | 29 (19–45) | 21 (14–32) | 25 (17–37) | 32 (22–47) | 33 (22.6–46.9) | 31 (21.6–44.8) |
| | HB (g/L) | 140 (135–146) | 139 (134–144) | 137 (133–141) | 143 (139–147) | 141 (137–145) | 145 (138–145) |
| | Iron (umol/L) | 21 (15–27) | 18 (13–24) | 17 (12–21) | 23 (19–27) | 22 (17–26) | 20 (15–24) |
| | TRF (g/L) | 3 (2–3) | 3 (2–3) | 3 (2–3) | 3 (2–3) | 3 (2–3) | 3 (2–3) |
| | TRS Sat (%) | 28 (20–39) | 23 (17–31) | 24 (19–31) | 31 (24–39) | 28 (22–36) | 26 (20–33) |
| **Aerobic Capacity** | $\dot{V}O_{2max}$ (ml kg$^{-1}$ min$^{-1}$) | 40*^# (36–44) | 45 (42–49) | 49 (46–52) | 41 (38–45) | 42 (39–45) | 45 (42–48) |
| **Force Production** | Isometric Mid-Thigh Pull (n) | 2175 (2029–2322) | 2180 (2037–2323) | 2106 (1976–2237) | 2155 (2032–2278) | 2212 (2091–2335) | 2241 (2121–2361) |
| | Isometric Bench Press (n) | - | 822^ (668–978) | 799^ (655–925) | - | 1024 (987–1151) | 999 (874–1123) |
| | Isometric Prone Row (n) | - | 628 (553–702) | 625 (553–697) | - | 678 (611–745) | 667 (601–733) |

Significance: p = ≤0.05,

*significantly different to testing point 2,

#significantly different to testing point 3,

^significantly different to forwards.

Kg—Kilograms, g—grams, %—percentage, n—newtons, N, ml kg$^{-1}$ min-1 —Millilitres per kilogram per minute, HB—haemoglobin, TRF—transferrin, TRS Sat—transferrin saturation

2 (p = 0.25), or 3 (p = 0.95). There was no significant difference in RMR within forwards or backs across testing points (backs: p ≥ 0.22, forwards: p ≥ 0.11; Table 5).

## Blood markers

Mean ferritin (backs: 21–28ug/L; forwards: 31–33ug/L), haemoglobin (backs: 137–140g/L; forwards: 141–145g/L), iron (backs: 17–21umol/L; forwards: 20–23umol/L), transferrin (backs: 3g/L; forwards: 3g/L), and transferrin saturation (backs: 23–28%; forwards: 26–31%), did not significantly differ between forwards and backs (p = ≥ 0.11) and did not significantly change across testing points (p ≥ 0.07) (Table 5).

## Aerobic capacity

The mean $\dot{V}O_{2max}$ for backs displayed a significant increase, rising from 40ml kg$^{-1}$ min$^{-1}$ at testing point 1 to 45ml kg$^{-1}$ min$^{-1}$ at testing point 2 and further to 49ml kg$^{-1}$ min$^{-1}$ at testing point 3 (p ≤ 0.04). The mean $\dot{V}O_{2max}$ for forwards increased from 41ml kg$^{-1}$ min$^{-1}$ at testing point 1 to 42ml kg$^{-1}$ min$^{-1}$ at testing point 2 and to 45ml kg$^{-1}$ min$^{-1}$ at testing point 3 however these changes were not significant (p ≥ 0.05). $\dot{V}O_{2max}$ did not differ between forwards and backs (p ≥ 0.07) (Table 5).

## Isometric force production

The mean isometric mid-thigh pull scores showed no significant differences between backs (2106–2180 N) and forwards (2155–2241 N) (p ≥ 0.13). Forwards exhibited greater isometric bench press scores at testing points 2 (forwards: 1024 N; backs: 822 N) and 3 (forwards: 999 N; backs: 799 N) (p ≤ 0.05). There were no significant differences in isometric prone row scores between backs and forwards (backs: 625–628 N; forwards 667–678 N) (p ≥ 0.31). Strength scores did not significantly change across testing points (p ≥ 0.09) (Table 5).

## Discussion

This is the first study to provide longitudinal changes specific to player positions for a range of anthropometric, physiological, and physical characteristics in international women's rugby league players. Notably, increases in lean mass were observed during the pre-season (from testing point 1 to testing point 2), and backs demonstrated improvements in $\dot{V}O_{2max}$ across the three testing points. Conversely, no significant changes were observed in fat mass, RMR, blood markers, or strength over time. Importantly, the call to better support female rugby league players through research [11] is addressed in this study. The novel longitudinal data presented supplements the existing evidence base and provides important information which may be used to profile and better support international women's rugby league players in practice.

Consistent with previous literature in women's rugby league players, this study confirms that forwards tend to have greater body mass and body fat percentages compared to backs, with no significant differences in lean mass [7–9]. Interestingly, both backs and forwards exhibited an increase in lean mass from testing point 1 to testing point 2, aligning with the training programme's emphasis on muscular hypertrophy during this phase. Previous research has shown increased training volume to enhance muscular hypertrophy [27]. The pre-season period in this study (testing point 1 to testing point 2) was characterised by increased training frequency and greater resistance training volume in comparison to the in-season period (testing point 2 to testing point 3). Therefore, the increase in lean mass at testing point 2 may be due to the increased resistance training volume which occurred throughout the pre-season period (Tables 2–4).

Despite the increase in lean mass, there were no significant changes observed in body mass or fat mass across the three testing points. It is worth noting that, when compared to data from 2014 [8], the average lean mass of English international backs increased by 3.8 kg, and forwards by 1.6 kg. In contrast, Australian backs (53.3 kg) and forwards (60.1 kg) have greater fat-free mass and a lower fat mass percentage (backs: 20.3%, forwards: 23.9%) than their English counterparts. We acknowledge that differences in positional classifications, such as the categorisation of hookers, locks, and half-backs, may have influenced these comparisons. However, given the recognised positive association between body mass, sprint momentum and collision dominance in female rugby [6] it is imperative to focus on developing the lean mass of international women's rugby league players to improve their performance.

This study provides a criterion assessed RMR data for women's senior international rugby players for the first time. Measured RMR closely align with the mean values reported for a combined sample of female elite (n = 18) and sub-elite (n = 18) rugby players in Ireland [28], differing by only 0.16 MJ.day$^{-1}$. However, it is noteworthy that the RMR values observed in this study are ~1.06 MJ.day$^{-1}$ and ~3.24 MJ.day$^{-1}$ lower than those reported for male senior professional rugby league and rugby union players in the English Super League [29] and Premiership [30], respectively. Furthermore, there were no significant differences in RMR across player positions (forwards vs. backs; 0.07 ± 0.87 MJ.day$^{-1}$) or testing points (-0.13 ± 1.03 MJ.day$^{-1}$).

Based on doubly labelled water-assessed energy expenditure data from our own laboratory, senior women's international rugby union players have a mean in-season physical activity level of 2.0 arbitrary units (AU) (range: 1.6–2.3 AU; Wilson et al., *unpublished observations*). When considering the mean RMR of 6.74 MJ.day$^{-1}$ measured in this study, the calculated total energy expenditure for a 73 kg women's international rugby player is ~13.48 MJ.day$^{-1}$ (range: 10.78–15.50 MJ.day$^{-1}$; 73 kg is the average body mass of players in this study). Following established sport nutrition guidelines (protein: 1.6–2.0 g.BM; fat: 20–35% of total energy expenditure) [31], this would allow for a daily dietary carbohydrate intake of 6.33–8.84 MJ.day$^{-1}$ to achieve energy balance (equivalent to 378–528 g or 5.2–7.2 g.BM of carbohydrate). Notably, female rugby players will need to consistently consume in excess of these values to achieve a positive energy balance associated with targeted increases in lean mass or overall body mass [31]. These findings now provide a valuable evidence base to support improved dietary intakes for female rugby players.

While iron profiles represented by blood measures did not show any statistical differences between player positions or testing points, it is notable that the mean ferritin levels remained consistently below the threshold for stage one iron deficiency (<35 ug/L) across all testing points [32]. Although not statistically significant, there was a trend towards decreases in ferritin levels from testing point 1 to testing point 2, particularly amongst backs. This trend was further emphasised by the classification of eleven players as stage 2 iron-deficient non-anaemia (Ferritin <20 ug/L) at different testing points. The observed changes in ferritin levels may be indicative of the increasing intensity of the training program and of the modest increases in lean mass observed during the pre-season period, a pattern consistent with the changes seen in female combat trainees over a 16 week basic combat training program [33]. Ferritin, unlike serum iron and transferrin saturation, tends to remain stable throughout the menstrual cycle [34], suggesting its specific role in capturing specific changes due to training load.

All players reported eumenorrhea at testing point 2, with a third (7 players) using hormonal contraceptives. While low iron stores, as approximated from ferritin levels, may compromise aerobic capacity or adaptations to training, the impact on lean mass remains unclear [35, 36]. These findings advocate for the inclusion of full iron profiles, rather than relying solely on haemoglobin levels, in the performance program for female rugby players. The evidence strongly

supports iron supplementation once ferritin levels drop below 20 ug/L [37], and dietary support to meet iron intakes requirements should be considered for players during intense training periods such as pre-season in this study.

Measures of force production, including isometric mid-thigh pull and prone row, showed no significant differences between forwards and backs, while forwards demonstrated greater bench press peak force. Peak force remained stable across the three testing points. Research in elite male rugby league players highlighted that pre-season improvements in lower body strength were linked to lifting heavier loads (>80% 1RM;) [38]. In contrast, this study's pre-season training emphasized muscular hypertrophy with lighter loads (8–12 repetitions at 70–80% 1RM), possibly explaining why isometric mid-thigh pull peak force did not increase alongside lean mass. Rugby league's multifaceted demands necessitate simultaneous development of various physical characteristics (e.g., strength, speed, and endurance), but concurrent training across modalities can sometimes limit strength gains [39]. Here, isometric peak force improvements were largely maintained rather than improved during the in-season period, in line with prior findings [38]. Comparing these results with previous research, the mean isometric mid-thigh pull peak force for Australian women's backs (+369 N) and forwards (+515 N) surpassed the values found for backs and forwards in this study [9]. Maximal strength is a pivotal determinant of successful rugby league performance, correlating with improved technical skill, greater match-play distances covered, enhanced sprint performance, and higher playing standards [38, 40]. Hence, prioritising maximal strength development during the pre-season period is crucial, as this phase often yields more noticeable strength improvements compared to the in-season phase [38].

Aerobic capacity analysis revealed no significant differences between backs and forwards, but a large disparity in their $\dot{V}O_{2max}$ improvements (backs: +22.5% forwards: +10%). It's important to note that a scarcity of research in female sports, particularly those studying longitudinal changes, hinders a direct comparison to the $\dot{V}O_{2max}$ values documented in this study. Minahan et al., [9] estimated the $\dot{V}O_{2max}$ of the Australian RL squad using a prediction equation based on their 30:15 intermittent fitness test score. The average $\dot{V}O_{2max}$ for English backs (+1.8 ml kg$^{-1}$ min$^{-1}$) and forwards (+2.1 ml kg$^{-1}$ min$^{-1}$) surpassed that of their Australian counterparts. Given the demands of rugby league match-play, where players must manage repeated high-intensity efforts and prolonged periods of play, a well-developed aerobic capacity is essential [3, 41]. This study shows that aerobic capacity can be developed during both pre-season and in-season periods, however, it may come at the cost of restricting the concurrent development of other critical anthropometric and physical qualities, such as lean mass and muscular strength.

## Limitations and strengths

Participants in this study were selected to be part of an international Rugby League World Cup squad, limiting recruitment to 24 participants. While the participant recruitment criteria ensured a high-quality sample of international standard athletes, it also imposes constraints on the study's statistical power and the generalisability of findings to the broader women's rugby league population. Moreover, it is important to acknowledge the absence of dynamic strength-power measures (e.g., 1-repetition maximum in the squat, vertical jumps) and sprint tests. Despite these limitations, which are common in applied research [42], this study also has several strengths. For example, this study utilised several criterion methods (i.e., indirect calorimetry and lab based $\dot{V}O_{2max}$ testing), while also evaluating critical elements of female athlete health (e.g., energy requirements and micronutrient status). As such, practitioners now have an enhanced evidence base to support athlete-centred interventions within practice.

## Conclusion

During the pre-season period, both forwards and backs exhibited increases in lean mass, while backs demonstrated significant improvements in $\dot{V}O_2$max across all testing points. However, no significant changes were observed in body mass, fat mass, body fat percentage, energy requirements, blood markers, or strength measures. The findings of this study contribute valuable position-specific normative data encompassing body composition, energy requirements, blood profiles, strength, and aerobic capacity for senior female rugby league players. These insights fill critical gaps in the existing evidence base and provide practitioners with knowledge to inform training strategies and better prepare women's rugby league players for the unique demands of their sport.

## Practical applications

The findings of this study offer practitioners normative values when assessing the unique characteristics of women's rugby league players, enabling a more precise and individualised approach to training and preparation. By analysing anthropometric, physiological, and physical traits, practitioners can identify position-specific strengths and weaknesses, providing insights for tailored training programs that align with the demands of women's rugby league. Furthermore, the inclusion of iron profile monitoring, especially during intensified training phases, can contribute to players' overall health and performance. The proposed energy and macronutrient reference values (energy: 0.14–21 MJ.kg$^{-1}$; carbohydrate: 5.2–7.2 g.BM; protein: 1.6 g.BM; fat 20–35% of total energy expenditure), equips practitioners with the means to optimise players' nutritional strategies, enhance body composition, and better prepare them for training and competition.

## Supporting information

**S1 Raw data. Data for analysis.**
(XLSX)

## Acknowledgments

The authors would like to thank the Rugby Football League and the England Performance Unit for supporting this study.

## Author Contributions

**Conceptualization:** Sarah Chantler, Carlos Ramirez, Peter Mackreth, Stuart Barrow, Ben Jones.

**Data curation:** Sean Scantlebury, Nessan Costello, Sarah Chantler, Carlos Ramirez, Neil Collins, Hayden Allen, Gemma Phillips, Marina Alexander, Matthew Barlow, Emily Williams, Peter Mackreth, Stuart Barrow, Parag Parelkar, Anthony Clarke, Benjamin Samuels, Stephanie Roe, Cameron Blake, Ben Jones.

**Formal analysis:** Sean Scantlebury, Nessan Costello, Cameron Owen.

**Methodology:** Nessan Costello, Cameron Owen, Hayden Allen, Ben Jones.

**Supervision:** Sean Scantlebury, Ben Jones.

**Writing – original draft:** Sean Scantlebury, Nessan Costello, Cameron Owen, Sarah Chantler, Santiago Zabaloy, Neil Collins, Hayden Allen, Gemma Phillips, Marina Alexander, Matthew Barlow, Ben Jones.

**Writing – review & editing:** Sean Scantlebury, Nessan Costello, Cameron Owen, Sarah Chantler, Carlos Ramirez, Santiago Zabaloy, Neil Collins, Hayden Allen, Emily Williams, Peter Mackreth, Stuart Barrow, Anthony Clarke, Benjamin Samuels, Stephanie Roe, Cameron Blake, Ben Jones.

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
