## [Decision Letter · Decision Letter 0]

10 Mar 2024

PONE-D-24-01874Longitudinal changes in anthropometric, physiological, and physical qualities of international women’s rugby league players.PLOS ONE

Dear Dr. Scantlebury,

Thank you for submitting your manuscript to PLOS ONE. After careful consideration, we feel that it has merit but does not fully meet PLOS ONE’s publication criteria as it currently stands. Therefore, we invite you to submit a revised version of the manuscript that addresses the points raised during the review process.

We look forward to receiving your revised manuscript.

Kind regards,

Jovan Gardasevic

Academic Editor

PLOS ONE

“I have read the journal's policy and the authors of this manuscript have the following competing interests:

SS is the strength and conditioning coach for England women's rugby league

SC is the nutritionist for England women's rugby league

NC is the sports scientist for England women's rugby league

GP is the head of medical for England women's rugby league

SB is the manager of England women's rugby league.

BJ is the head of performance for England rugby league”

3. We note that your Data Availability Statement is currently as follows: [All relevant data are within the manuscript and its Supporting Information files]

Reviewers' comments:

Reviewer's Responses to Questions

**Comments to the Author**

1. Is the manuscript technically sound, and do the data support the conclusions?

Reviewer #1: Partly

Reviewer #2: Yes

2. Has the statistical analysis been performed appropriately and rigorously? 

Reviewer #1: Yes

Reviewer #2: Yes

3. Have the authors made all data underlying the findings in their manuscript fully available?

Reviewer #1: Yes

Reviewer #2: Yes

4. Is the manuscript presented in an intelligible fashion and written in standard English?

Reviewer #1: Yes

Reviewer #2: Yes

5. Review Comments to the Author

Reviewer #1: A very interesting study. It is particularly interesting because we have a longitudinal study, which is very rare today.

Methodologically, the paper is very well laid out. Language and writing style also very good. What I can suggest to pay particular attention to and correct is:

- We must adhere to the rules and principles of writing throughout the work. Namely, in one situation you write the title of the table (Table 1) below the table itself, and in the next situation you write the title of the table (Table 2) above the table. This must be uniform.

- Paragraph 251 (it seems to me that it is not written in the same font and size as the other paragraphs)

- Display of results very well done

- The discussion should be more detailed. Namely, very often you state some data from the results, point out that it is interesting and that's where the explanation ends. It is very important to note why there is or is not statistical significance somewhere. Of course, the explanation can be your thought, or a conclusion based on some previous research. In this way, the discussion remains deprived of some clarifications. This refers to the first part of the discussion up to paragraph 351. From paragraph 351 until the end of the discussion, I have no objections.

Reviewer #2: First of all, I find the work interesting. the fact that the first longitudinal work is related to the issue of work gives additional value to the work, because much more data is obtained through longitudinal research. The paper is generally good, which means that all the methodological rules of writing the paper have been followed, from the introduction in which the authors introduce the reader to all terms related to the research problem. In doing so, but also in the rest of the work, they use a large number of references that are fairly recent, which is commendable. The choice of respondents, instruments that were used is correct. The tables clearly and concretely describe the microcycles and training programs in the preseason and the season itself. What also gives the work value is the large number of data obtained by measurement. All these data were correctly interpreted with an adequate statistical method, through a rich discussion, and finally the correct conclusions were drawn. The authors also clearly pointed out the advantages and disadvantages of the work itself. I have no major complaints. I would like to ask the authors about the breakfast they received during the research, what is meant by the term standard breakfast and whether they made a targeted selection of it, i.e. proteins and carbohydrates and other components. The second question relates to the off-season period, whether the authors believe that two to three training sessions are enough for further progress, given that they are top athletes. And finally, a technical comment. The title of Table 1 should be above the table and not below it. In any case, I believe that the work deserves to be published in a journal.

6. PLOS authors have the option to publish the peer review history of their article (what does this mean?). If published, this will include your full peer review and any attached files.

Reviewer #1: No

Reviewer #2: No

---

## [Author Response · Author response to Decision Letter 0]

4 Apr 2024

Thank you very much for your review of the manuscript. We have amended the discussion based on your suggestions. We hope you find the changes suitable. 

Reviewer #1: A very interesting study. It is particularly interesting because we have a longitudinal study, which is very rare today. Methodologically, the paper is very well laid out. Language and writing style also very good. What I can suggest to pay particular attention to and correct is:

We must adhere to the rules and principles of writing throughout the work. Namely, in one situation you write the title of the table (Table 1) below the table itself, and in the next situation you write the title of the table (Table 2) above the table. This must be uniform.

Thank you for pointing this out, this appears to have occurred when uploading the manuscript to PLOS one. This has now been corrected. 

Paragraph 251 (it seems to me that it is not written in the same font and size as the other paragraphs)

This may be a technical issue with the manuscript upload. We have made sure all text is now Times New Roman size 12.

Display of results very well done

The discussion should be more detailed. Namely, very often you state some data from the results, point out that it is interesting and that's where the explanation ends. It is very important to note why there is or is not statistical significance somewhere. Of course, the explanation can be your thought, or a conclusion based on some previous research. In this way, the discussion remains deprived of some clarifications. This refers to the first part of the discussion up to paragraph 351. From paragraph 351 until the end of the discussion, I have no objections.

Thank you for your comment, we agree that some elements of the discussion would benefit from additional detail. Whilst the first paragraph remains an overview of the main findings, greater explanation has been added into the paragraphs up to line 351.

Thank you very much for taking the time to review the manuscript and thank you for your comments. We have addressed your comments below, we hope you find the responses suitable.

Reviewer #2: First of all, I find the work interesting. the fact that the first longitudinal work is related to the issue of work gives additional value to the work, because much more data is obtained through longitudinal research. The paper is generally good, which means that all the methodological rules of writing the paper have been followed, from the introduction in which the authors introduce the reader to all terms related to the research problem. In doing so, but also in the rest of the work, they use a large number of references that are fairly recent, which is commendable. The choice of respondents, instruments that were used is correct. The tables clearly and concretely describe the microcycles and training programs in the preseason and the season itself. What also gives the work value is the large number of data obtained by measurement. All these data were correctly interpreted with an adequate statistical method, through a rich discussion, and finally the correct conclusions were drawn. The authors also clearly pointed out the advantages and disadvantages of the work itself. I have no major complaints. 

I would like to ask the authors about the breakfast they received during the research, what is meant by the term standard breakfast and whether they made a targeted selection of it, i.e. proteins and carbohydrates and other components. 

The breakfast was standardised so that all athletes received the same breakfast meal prior to their remaining physical testing (They would need to be fasted for DXA and RMR measures). The targeted selection was to ensure a meal that was familiar as well as nutritionally balanced with a reasonable energy density and carbohydrate volume. 

The second question relates to the off-season period, whether the authors believe that two to three training sessions are enough for further progress, given that they are top athletes. 

Thank you for your comment. We agree that 2-3 sessions per week is not sufficient training volume to provide a stimulus for adaptation for elite athletes. However, as this was the off-season period the focus was on physical and mental recovery for 3 months, following the previous year’s competition. Therefore, training volume was reduced to try and provide the athletes sufficient opportunity to recover whilst also minimising the effects of de-training. Following the off-season period, training frequency increased to 4-5 sessions per week in the pre-season period with greater session volume and intensity. 

And finally, a technical comment. The title of Table 1 should be above the table and not below it. In any case, I believe that the work deserves to be published in a journal.

Thank you for pointing this out, this appears to have occurred during the upload to the journal. This has now been corrected.

---

## [Decision Letter · Decision Letter 1]

12 Apr 2024

Longitudinal changes in anthropometric, physiological, and physical qualities of international women’s rugby league players.

PONE-D-24-01874R1

Dear Dr Sean Scantlebury,

We’re pleased to inform you that your manuscript has been judged scientifically suitable for publication and will be formally accepted for publication once it meets all outstanding technical requirements.

Kind regards,

Jovan Gardasevic

Academic Editor

PLOS ONE

Additional Editor Comments (optional):

Dear Author,

I read the manuscript and the reviewers' comments in detail. In my opinion, the manuscript is now ready for publication in this respectable journal.

Kind regards,

reviewer

Reviewers' comments:

Reviewer's Responses to Questions

**Comments to the Author**

1. If the authors have adequately addressed your comments raised in a previous round of review and you feel that this manuscript is now acceptable for publication, you may indicate that here to bypass the “Comments to the Author” section, enter your conflict of interest statement in the “Confidential to Editor” section, and submit your "Accept" recommendation.

Reviewer #1: (No Response)

2. Is the manuscript technically sound, and do the data support the conclusions?

Reviewer #1: Yes

3. Has the statistical analysis been performed appropriately and rigorously? 

Reviewer #1: Yes

4. Have the authors made all data underlying the findings in their manuscript fully available?

Reviewer #1: Yes

5. Is the manuscript presented in an intelligible fashion and written in standard English?

Reviewer #1: Yes

6. Review Comments to the Author

Reviewer #1: the authors made an effort to accept and correct everything related to my suggestions, so that the paper is now very well done.

7. PLOS authors have the option to publish the peer review history of their article (what does this mean?). If published, this will include your full peer review and any attached files.

Reviewer #1: No

---

## [Editor Report · Acceptance letter]

29 Apr 2024

PONE-D-24-01874R1 

PLOS ONE

Dear Dr. Scantlebury, 

I'm pleased to inform you that your manuscript has been deemed suitable for publication in PLOS ONE. Congratulations! Your manuscript is now being handed over to our production team.

Kind regards, 

on behalf of

Dr. Jovan Gardasevic 

Academic Editor

PLOS ONE